

# Optimization of aeration enhanced surfactant soil washing for remediation of diesel-contaminated soils using response surface methodology

Befkadu Abayneh Ayele[1,2], Jun Lu[1] and Quanyuan Chen[1,3]

[1] State Environmental Protection Engineering Center for Pollution Treatment and Control in Textile Industry, Donghua University, Shanghai, P.R. China
[2] Department of Natural Resource Management, University of Gondar, Gondar, Ethiopia
[3] Shanghai Institution of Pollution Control and Ecological Security, Shanghai, P.R. China

## ABSTRACT

Surfactant-enhanced soil washing has been used for remediation of organic pollutants for an extended period, but its effectiveness and wide application was limited by the high concentration of surfactants utilized. In this work, the efficiency of conventional soil washing performance was enhanced by 12–25% through the incorporation of air bubbles into the low concentration surfactant soil washing system. Surfactant selection pre-experiment using aerated and conventional soil washing reveals Brij 35 > TX100 > Tween 80 > Saponin in diesel oil removal. Optimization of the effect of time, surfactant concentration, pH, agitation speed, and airflow rate in five levels were undertaken using Response Surface Methodology and Central composite design. The optimum degree of variables achieved was 90 min of washing time, 370 mg/l of concentration, washing pH of 10,535 rpm of agitation speed and 7.2 l/min of airflow rate with 79.5% diesel removal. The high predicted $R^2$ value of 0.9517 showed that the model could efficiently be used to predict diesel removal efficiency. The variation in efficiency of aeration assisted and conventional soil washing was variable depending on the type of surfactant, organic matter content of the soil, particle size distribution and level of pollutant weathering. The difference in removal efficiency of the two methods increases when the level of organic matter increases and when the particle size and age of contamination decreases.

## INTRODUCTION

Soil contamination by petroleum hydrocarbons is a major concern for environmentalists as hydrocarbons have a strong affinity to organic matter present in the soil (*Wang et al., 2019b*), which pauses potential risks to the water resources, ecosystem, human health and other environmental receptors (*Obida et al., 2018*; *Streche et al., 2018*). Petroleum products contaminate soils from different leaks and spills from refineries, manufacturing sites, power plants, distribution depots, retail service stations, land disposal sites and

Corresponding author
Quanyuan Chen,
qychen@dhu.edu.cn

corroded pipelines (*Al-Hawash, Zhang & Ma, 2019*; *Buzmakov & Khotyanovskaya, 2020*; *Hernández-Espriú et al., 2013*). Therefore, economical and efficient remediation of contaminated soils is in pressing need for proper ecosystem functioning (*Kim et al., 2019*).

Different physical, chemical, and biological remediation techniques are commonly utilized to treat the petroleum polluted soils (*Bushnaf et al., 2017*; *Guarino et al., 2019*; *Luo et al., 2019a*); these methods aim at separating, oxidizing, modifying pollutants into harmless byproducts and reducing contaminant concentration (*Ali et al., 2019*; *Kim et al., 2019*; *Streche et al., 2018*; *Wang et al., 2019b*). Soil washing using environmentally friendly surfactants has broader acceptance as it is an efficient, simple, fast, low cost, ecologically friendly technique and applies to a broader category of contaminant treatment (*Gusiatin & Radziemska, 2018*; *Mao et al., 2015*; *Mousset et al., 2014a*).

Soil washing using surfactants transfers the pollutants from the soil colloid to the soil solution due to two central mechanisms called mobilization and solubilization (*Alzahid et al., 2019*; *Javanbakht & Goual, 2016*). At low concentration, the surfactant molecules exist as monomers, but when their concentration in the solution increases up to their critical micelle concentration (CMC), they start forming micelles (*Phukon, Nandi & Sahu, 2018*). Micelles contain a hydrophobic interior that entraps the pollutants inside and a hydrophilic exterior that faces the aqueous solution; through this mechanism, they reduce the surface tension of the system (*Elgh-Dalgren et al., 2009*). The reduction in surface tension after CMC mobilizes and solubilizes contaminants from the surface of soil colloids to the soil solution (*Belhaj et al., 2019*).

Among the different surfactant groups, anionic and nonionic surfactants are commonly used for remediation of petroleum polluted soils, the lower CMC and lower precipitation properties of nonionic surfactants makes them more preferable (*Befkadu & Chen, 2018*). For this study, we selected four commonly used nonionic surfactants TX100, Tween 80, Brij 35 and Saponin for diesel oil removal. From pre-experiment results conducted on highly diesel contaminated soils, Brij 35 surfactant was selected for further optimization study due to its effectiveness in diesel removal.

Many of the researches conducted on surfactant enhanced soil washing used a very high concentration of surfactants up to six g/l to remove the contaminants (*Dos Santos et al., 2017*; *Pei et al., 2018*). Using such a high concentration of surfactants on treating petroleum contaminated soils, have economic and ecological effects as a stable emulsion will be formed between pollutants and fine soil particles (*Trellu et al., 2016*). If we use low surfactant concentration for treatment, the concentration of surfactant in the washing effluent will be lower, and it will interfere less with advanced oxidation processes applied for washing solution treatment and oil–water separation (*Mousset et al., 2014b*). Finding technologies that can reduce the concentration of surfactant used to a lower level will simplify post-treatment loads, and selective absorption works, as there are few surfactants to recover.

In this work, we develop a remediation method that incorporates air bubbles into the soil washing system, which achieves improved diesel removal than conventional soil

Table 1 Physicochemical properties of surfactants used in the research.

| Surfactant | Molecular formula | HLB | Molecular weight | CMC (gl-1 25 °C) |
|---|---|---|---|---|
| Brij 35 | $C_{12}H_{25}(OC_2H_4)_{23}OH$ | 16.9 | 1,198 | 0.2 |
| Tween 80 | $C_{64}H_{124}O_{26}$ | 15 | 1,309 | 0.015 |
| Tx-100 | $C_{34}H_{62}O_{11}$ | 13.4 | 624 | 0.15 |
| Saponin | – | 16 | – | 0.14 |

washing under lower surfactant concentration. Air sparging is commonly used for in situ soil remediation to remove volatile hydrocarbons (*Mousset et al., 2014a*), but there is no reported work on using it in ex situ surfactant enhanced soil washing. In addition, we optimize and validate the soil washing conditions by using response surface methodology (RSM) with central composite design (CCD), which is necessary to scale up the lab-based soil washing tests, reduce cost and improve the efficiency of contaminant removal (*Asadzadeh et al., 2018*).

Response surface methodology requires a smaller number of runs, and detects interactions, models, and predicts the washing process and performance better than the commonly utilized one factor at a time optimization approach (*Montgomery, 2013*). Five operating factors surfactant concentration, washing time, agitation speed, soil solution pH and airflow rate are selected and optimized. Soil properties like soil texture, organic matter content, mineral composition of the soil and pH significantly affect its contaminant desorption rate (*Gusiatin & Radziemska, 2018*). We also validate and compared the optimized operating factors under different soil textural classes, organic matter levels and age of contamination between aerated and non-aerated soil washing conditions. The improved performance of aerated enhanced soil washing method developed in this research is expected to serve as a starting point for further field trial studies.

# MATERIALS AND METHODS

## Chemicals and reagents

Analytical quality Surfactants: Brij 35, Tween 80, TX-100, and Saponin (98% purity) and anhydrous sodium sulfate, $K_2Cr_2O_7$ and $FeSO_4$ were obtained from Sinopharm Chemical Reagent Co., Ltd., China and were used as received. Table 1 shows the physicochemical properties of the surfactants used in this research.

## Soil sampling, analysis and contamination

The soil sample was collected from Mr. Zheng Sun's farmland upon his approval from top 20 cm of the land with no history of petroleum contamination from Songjiang district, Shanghai, China. The soil sample was air-dried at room temperature, tapped to break aggregated soils then homogenized and sieved to remove large sand, gravels and plant roots having a size of more than 20 mesh. The particle size distribution of the soil was analyzed with BT-9300S Laser Particle Size Analyzer, organic matter content of the soil was analyzed with the Walkley–Black procedure, and soil mineralogy was analyzed with X-ray diffraction. A fixed mass of sieved soil was contaminated with Diesel oil obtained from gasoline station at room temperature and was brought to a contamination

**Table 2 The physico-chemical properties of the soil.**

| OM % | pH in water | Particle size in μm | | | | | | Textural class |
|------|-------------|------|------|--------|---------|---------|---------|----------------|
|      |             | <2   | 2-50 | 50–150 | 150–250 | 250–500 | 500–841 | Silt Loam |
| 5.4  | 8.2         | 10.6 | 73.3 | 15.7   | 0.4     | –       | –       |  |

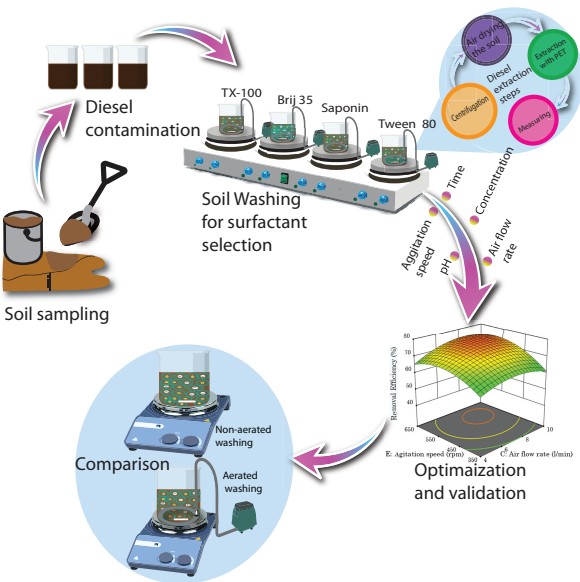

**Figure 1 Schematic illustration from soil sampling, artificial contamination, surfactant selection, optimization to method comparison.**

level of 10%. An air-dried soil was weighted, a quantified amount of diesel oil dissolved with petroleum ether was added to the soil, and the soil was thoroughly mixed to get uniform contamination. Contaminated soils were stored for one year in the hood uncovered, which allow the petroleum ether and light diesel fractions to evaporate which helped to mimic the natural weathering conditions. The physicochemical properties of the sampled soil used for this research are presented in Table 2.

## General soil washing experiment

Twenty grams of artificially diesel-contaminated soil was mixed with 400 ml of surfactant solutions and were added to a 1,000 ml beaker magnetic stirrer bars and aerators were also added to the beaker. The soil-washing scheme from soil sampling to contamination then surfactant selection and finally method comparison is shown in Fig. 1. The contaminated soil was washed with different types and concentrations of surfactants, soil solution pH, stirring speed, washing time and airflow rate according to the design of the experiment. Soil/liquid ratio of 1:20 was chosen from preliminary experimental results, and this ratio was fixed throughout all experiments to reduce the number of parameters to be optimized. Aeration was supplied to the washing solution with circular aerator of diameter 20 cm using aeration pumps as shown in Fig. 1 (CX-0088 type oxygen pump; Chuangxing Electrical Appliance Co., Ltd.,

Zhongshan China) and the airflow rate was measured with LZB-4 Wb type rotor flowmeter. After the soil is washed for the intended time, the soil and the washing solution were separated by centrifuging (R centrifuge: Thermo Fisher Science and Technology Limited Company, Waltham, MA USA) the mixture at 3,500 rpm for 15 min. after centrifuge the soil was rinsed off with distilled water and will be left open for air drying.

## Washing efficiency comparison

To identify the effectiveness of aeration-assisted soil washing under different soil organic matter levels, the age of contamination and textural classes, a comparative study of aerated assisted soil washing and regular soil washing was made under the optimized washing conditions. 1%, 3%, 5% and 7% of humus were added to the soil to modify the organic matter content of the soil proportionally. For the aging experiment, the soils were contaminated for 1, 3, 6, 9 and 12 months. In the experiment investigating different types of soil textural classes, the contaminated soil was sieved into four different size classes, and the diesel removal efficiency of surfactant-enhanced soil is washing with aeration and without aeration was compared in those classes. The diesel removal efficiency, was calculated using Eq. (1) and all experiments were undertaken in three replicates.

$$\text{Diesel removal} \, (\%) = \frac{C_o - C}{C_o} * 100 \tag{1}$$

where $C_o$ is the initial diesel concentration in the artificially polluted soil and $C$ is the concentration of diesel remaining in the washed soil.

## Diesel extraction and analysis

Two grams of washed and air-dried soil samples mixed with five gm anhydrous sodium sulfate were solvent-extracted with 10 ml of petroleum ether in plastic-covered flasks in a horizontal shaker (COS-110X50 type water bath shaker: Shanghai Instruments Biofuge Primo Co. Ltd., Shanghai, China) for 15 min with a shaking speed of 250 rpm (*Sui et al., 2014*). The extraction procedure was repeated with fresh petroleum ether until the final extraction gives the same absorbance reading as raw petroleum ether. Three cycles of extraction cycles were enough to extract all the oil remaining in the soil. All supernatants from the three cycles of extraction were filtered and mixed, and petroleum ether was added to make a final extractant solution of 40 ml. The filtrate was diluted, and the diesel oil concentration in the dilute was determined by absorbance reading in UV spectrophotometer at 225 nm (*Lu et al., 2011*; *Wang et al., 2019a*; *Zheng et al., 2019*). Then the concentration of diesel was determined from the regression equation prepared from calibration of a known concentration and absorbance.

## Determination of surfactant CMC

The CMC values of the four surfactants were determined by applying the surface tension technique. Surface tensions of pure surfactant solutions over a wide concentration range were prepared and allowed to equilibrate for approximately 5 h before measurements

**Table 3 Experimental range and levels of independent variables.**

| Washing parameters | Coded variables | Unit | Range | Varied washing parameter levels | | | | |
|---|---|---|---|---|---|---|---|---|
| | | | | $-\alpha$ | $-1$ | 0 | 1 | $+\alpha$ |
| Time | A | min | 30–150 | 30 | 60 | 90 | 120 | 150 |
| pH | B | | 4–12 | 4 | 6 | 8 | 10 | 12 |
| Airflow rate | C | l/min | 2–10 | 2 | 4 | 6 | 8 | 10 |
| Concentration | D | mg/l | 100–500 | 100 | 200 | 300 | 400 | 500 |
| Agitation speed | E | rpm | 250–650 | 250 | 350 | 450 | 550 | 650 |

were made using a Kruss processor Tensiometer (BP-100 type bubble pressure type state surface tension instrument: German Kruss Company, Hamburg, Germany) (*Sales, De Rossi & Fernández, 2011*). When a steady reading is attained, the surface tension reading is measured, as shown by at least three consecutive readings with nearly identical values. Measurements were made until the increase in surfactant concentration did not cause a further decrease in the surface tension. By plotting the surface tension values against the logarithm of surfactant concentration, the CMC value could be found at a concentration where it shows the least surface tension (*Muherei & Junin, 2007*).

## Response surface methodology

Response surface methodology and CCD was used to model, analyze and optimize the impact of the independent variables on the response variable using mathematical and statistical techniques (*Castorani, Landi & Germani, 2016*). Five soil washing parameters, agitation speed, surfactant concentration, washing time, soil solution pH, and airflow rate were chosen as independent variables and diesel removal efficiency as the response variable. Each of the five independent variables was tested at five levels coded as $-\alpha$, $-1$, 0, and 1 and $\alpha$ respectively, the coded and actual values of the five parameters and their ranges are shown in Table 3. For statistical calculation, the independent variables are converted into coded values according to Eq. (2):

$$X_i = \frac{Z_i - Z_{io}}{\Delta Z_i} \tag{2}$$

where $X_i$ represents the coded value of the variable, $Z_i$ represents the experimental value of the variable, $Z_{io}$ the actual value of $Z_i$ at the center point; and $\Delta Z_i$ represents the step change value (*Xu et al., 2017*). The experiment was conducted according to CCD the determination of experimental runs and analysis of the results were undertaken using Design-Expert version 10 (Stat Ease Inc., Minneapolis, MN, USA). The total number of runs for half CCD was calculated by the equation $n = 2^{k-1} + 2k + X_0$ where $n$ is the total number of experiments, $k$ is the number of factors, $X_0$ is the number of central points (*Dong & Sartaj, 2016*; *Khoobbakht et al., 2016*). Thus, this CCD consists 16 of factorial points, 10 axial points and six center points that gave a total of 32 experimental runs, and the central point was used to estimate the pure error sum of squares. To find the optimum diesel removal efficiency and the optimal soil washing conditions experiments were

performed in three replicates in random order and the average of the obtained data was fitted into the second-order polynomial regression model Eq. (3).

$$Y = A_o + \sum_{i=1}^{n} A_i X_i + \sum_{i=1}^{n} A_{ii} X_i^2 + \sum_{i \neq 1}^{n} \sum_{j=1}^{n} A_{ij} X_i X_j + \varepsilon \qquad (3)$$

where $Y$ represent the predicted response variable (diesel removal efficiency), $A_o$ represents the value of the fixed response at the center point of the design; $A_i$, $A_{ii}$, $A_{ij}$ represents the linear, quadratic and interaction effects regression terms, respectively; $x_i$ denotes the level of the independent variable; $n$ is the number of independent variables, and $\varepsilon$ is the random error (*Montgomery, 2013*). The significance of each coefficient and the validity of the model is investigated with multiple regression analysis using analysis of variance (ANOVA) with *F*-test and lack of fit tests at 95% confidence level and *p*-values smaller than 0.05 were considered statistically significant. The adequacy of the model was expressed by the coefficient of determination ($R^2$).

## RESULTS AND DISCUSSION

### Prescreening of washing parameters and surfactants

The efficiency of ex-situ soil washing is affected by factors including washing time, soil-washing solution pH, surfactant concentration, agitation speed, soil to liquid ratio, washing temperature, on and off mode and number of extractions (*Peng, Wu & Chen, 2011*; *Zou et al., 2009*). Of these factors temperature and on and off mode are reported to be uneconomical and to have less impact, and we have conducted the experiments under room temperature, with one-time extraction, and with constant washing mode (*Chao, Hsieh & Tran, 2018*). For this research, we have included an airflow rate as an additional factor affecting washing efficiency and kept soil to liquid ratio constant at 1:20 from pre-experiment and reported ratios (*Peng, Wu & Chen, 2011*). Table 3 presents the experimental range and levels of soil washing parameters used in this optimization study.

Before the optimization experiment, a prescreening surfactant selection experiment was conducted using nonionic surfactants Brij 35, Tween 80, Triton-X100 and a nonionic biosurfactant Saponin. The surfactant selection experiment was conducted under the following operating conditions washing pH of 8.3, a surfactant concentration of 300 mg/l, agitation speed of 450 rpm, washing time of 1 h, and an airflow rate of 6 l/min. The results of the surfactant selection experiment in the order of diesel removal efficiency was Brij 35 > TX100 > Tween 80 > Saponin (65.6%, 61.5%, 58.7% and 41.7% respectively). Based on this pre-experiment Brij 35 was selected for further optimization study of soil washing operating conditions, *Zamudio-Pérez et al. (2013)* also compared 15 synthetic surfactants for remediation of petroleum hydrocarbons and found that Brij 35 surfactant to be the most efficient.

### Experimental design and RSM model optimization

RSM with CCD has been used for developing the mathematical relationship between the independent and the response of variables, for detecting the interaction factors and for
**Table 4 Experimental runs, predicted and experimental efficiency.**

| Run | Time (A)<br>min | pH (B)<br>– | Air flow rate (C)<br>l/min | Concentration (D)<br>mg/l | Agitation (E)<br>rpm | Mean experimental value<br>% | Standard deviation | Predicted value<br>% |
|---|---|---|---|---|---|---|---|---|
| 1 | 90(0) | 8(0) | 6(0) | 300(0) | 450(0) | 71.6 | 0.79 | 71.7 |
| 2 | 90(0) | 8(0) | 6(0) | 300(0) | 450(0) | 72.6 | 0.60 | 71.7 |
| 3 | 90(0) | 8(0) | 6(0) | 300(0) | 450(0) | 70 | 0.56 | 71.7 |
| 4 | 120(1) | 10(1) | 4(−1) | 200(−1) | 550(1) | 63.3 | 0.70 | 62.5 |
| 5 | 60(−1) | 10(1) | 4(−1) | 400(1) | 550(1) | 69 | 0.56 | 68.4 |
| 6 | 120(1) | 10(1) | 8(1) | 200(−1) | 350(−1) | 59.5 | 0.72 | 59.9 |
| 7 | 90(0) | 8(0) | 6(0) | 300(0) | 650(+α) | 70 | 0.68 | 71.7 |
| 8 | 90(0) | 8(0) | 6(0) | 300(0) | 450(0) | 71 | 0.46 | 71.7 |
| 9 | 90(0) | 12(+α) | 6(0) | 300(0) | 450(0) | 72.7 | 0.76 | 73.3 |
| 10 | 150(+α) | 8(0) | 6(0) | 300(0) | 450(0) | 69.1 | 0.67 | 69.3 |
| 11 | 90(0) | 8(0) | 2(−α) | 300(0) | 450(0) | 57.7 | 0.72 | 58.2 |
| 12 | 90(0) | 8(0) | 10(+α) | 300(0) | 450(0) | 70.9 | 0.67 | 70.3 |
| 13 | 30(−α) | 8(0) | 6(0) | 300(0) | 450(0) | 57 | 0.25 | 56.6 |
| 14 | 60(−1) | 10(1) | 4(−1) | 200(−1) | 350(−1) | 46.2 | 0.45 | 46.4 |
| 15 | 90(0) | 8(0) | 6(0) | 300(0) | 450(0) | 71.3 | 0.36 | 71.7 |
| 16 | 120(1) | 6(−1) | 8(1) | 400(1) | 350(−1) | 61.6 | 0.21 | 62.3 |
| 17 | 120(1) | 6(−1) | 4(−1) | 200(−1) | 350(−1) | 49.8 | 0.47 | 50.2 |
| 18 | 90(0) | 8(0) | 6(0) | 100(−α) | 450(0) | 50.4 | 0.20 | 50.5 |
| 19 | 60(−1) | 6(−1) | 8(1) | 400(1) | 550(1) | 64.8 | 0.68 | 64.8 |
| 20 | 60(−1) | 6(−1) | 4(−1) | 200(−1) | 550(1) | 52.9 | 0.67 | 52.6 |
| 21 | 60(−1) | 6(−1) | 4(−1) | 400(1) | 350(−1) | 50.3 | 0.10 | 50.9 |
| 22 | 120(1) | 6(−1) | 8(1) | 200(−1) | 550(1) | 66.9 | 0.57 | 66.7 |
| 23 | 90(0) | 8(0) | 6(0) | 300(0) | 450(0) | 73.8 | 0.38 | 71.7 |
| 24 | 120(1) | 10(1) | 4(−1) | 400(1) | 350(−1) | 65.3 | 0.49 | 65.5 |
| 25 | 120(1) | 10(1) | 8(1) | 400(1) | 550(1) | 79.5 | 0.47 | 79 |
| 26 | 90(0) | 8(0) | 6(0) | 500(+α) | 450(0) | 67.4 | 0.42 | 67.2 |
| 27 | 60(−1) | 10(1) | 8(1) | 400(1) | 350(−1) | 62.7 | 0.56 | 63.3 |
| 28 | 60(−1) | 10(1) | 8(1) | 200(−1) | 550(1) | 67.6 | 0.76 | 67.2 |
| 29 | 90(0) | 8(0) | 6(0) | 300(0) | 250(−α) | 53.3 | 0.44 | 51.5 |
| 30 | 120(1) | 6(−1) | 4(−1) | 400(1) | 550(1) | 64.1 | 0.31 | 63.7 |
| 31 | 90(0) | 4(−α) | 6(0) | 300(0) | 450(0) | 60.1 | 0.71 | 59.4 |
| 32 | 60(−1) | 6(−1) | 8(1) | 200(−1) | 350(−1) | 44.7 | 0.75 | 45.5 |

optimizing parameter values (*Tan et al., 2017*). Table 4 shows the CCD matrix generated by Design expert software and the diesel removal efficiency in experimental and predicted values. The following second-order mathematical model was developed for predicting diesel oil removal efficiencies from soil washing as a simultaneous function of washing time (A), pH (B), airflow rate (C), surfactant concentration (D) and agitation speed (E).

**Table 5  Model fitting results.**

| Model parameter | Full quadratic model | Enhanced model |
|---|---|---|
| $R^2$ | 0.9917 | 0.9875 |
| Adjusted $R^2$ | 0.9766 | 0.9772 |
| Predicted $R^2$ | 0.8671 | 0.9517 |
| $p$-Value for lack-of-fit | 0.46 | 0.5148 |

$$\text{Diesel removal } (\%) = 71.7 + 3.2 * A + 3.5 * B + 3.0 * C + 4.2 * D + 5.1 * E$$
$$- 0.5 * AB + 0.2 * AC - 0.3 * AD - 0.8 * AE + 0.3 * BC$$
$$+ 0.8 * BD + 0.2 * BE - 0.4 * CD + 0.8 * CE - 0.8 * DE$$
$$- 2.2 * A^2 - 1.4 * B^2 - 1.9 * C^2 - 3.2 * D^2 - 2.5 * E^2$$

Sequential analysis of the $F$-test, lack of fit test and ANOVA analysis are commonly used to check the consistency of the RSM model (*Ragavendran et al., 2017*). The purpose of ANOVA analysis is to investigate whether the soil washing parameters and their interactions have a significant effect on diesel removal efficiency and to identify the significance of the model. The analysis result presented in Table 5, indicates that the model has an $F$-value of 65.57 with a significant $p$-value of $< 0.0001$ indicating the adequacy of the model. The chance that this large $F$ value could occur due to noise is less than 0.01%; we can accept this model with 99.99% confidence level (*Montgomery, 2013*). ANOVA results revealed that all linear and quadratic model terms and four interaction terms (AE, BD, CE and DE) are significant factors affecting diesel removal with a $p$-value of $< 0.05$, with 95% confidence level. The significance of interaction terms justifies the hypothesis that optimization study should consider both the main factors and interaction terms to sufficiently explain the washing condition. The calculated lack of fit $F$-value of 1.12 is not significant compared to the pure error which is good, as we want the model to fit the response and there is a 46% chance that an $F$ value this large could happen due to noise (*Safari et al., 2018*).

The determination coefficient ($R^2$) represents a clear correlation between measured and predicted values, and an $R^2$ value of 0.9917 of this experiment implies that the model explains 99.17% of the sample variation from the total variation (*Montgomery, 2013*). Moreover, the adjusted $R^2$ is 0.9766, and the predicted $R^2$ is 0.8671, a reasonable variation among predicted and adjusted $R^2$ of 0.2 indicates the adequacy of the model (*Ng, Sen Gupta & Hashim, 2015*; *Nosrati, Jayakumar & Hashim, 2011*). Higher Predicted $R^2$ of 0.8671 tells the models higher predictive capability for new observations, a reasonably close fitting of predicted and experimental results are also shown in Fig. 2, which confirms the accuracy of the model. Adequate precision represents the signal to noise ratio and a value higher than four is desired (*Long, Zhang & Lei, 2013*). The ratio found here is 30.27 which implies adequate signal of this study. A lower coefficient of variation (CV) 2.16% of this analysis indicates the reliability of the experiment as the CV is less than 10% (*Martínez Álvarez et al., 2015*).
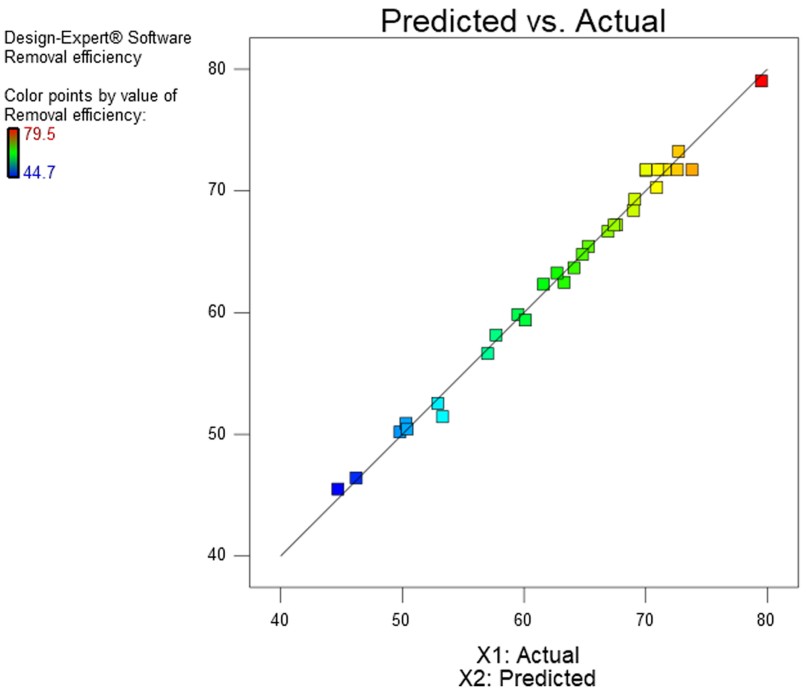

**Figure 2 Comparison plot of predicted and experimental diesel removal efficiencies.**

## Model improvement

Removing insignificant model terms from RSM models could potentially improve the predicting ability of the full quadratic models (*Dutka, Ditaranto & Løvås, 2015*). The predicting ability of Eq. (1) was enhanced by removing non-significant factors (one-by-one) with *p*-values higher than 0.05 by backward selection mode. The improved model regression analysis for diesel removal efficiency was based on five linear and quadratic effects, and four two-way interaction effects, the other six non-significant interaction effects were not considered in the enhanced model. Although the $R^2$ and Adjusted $R^2$ values of the improved model were slightly lower than the full quadratic model as indicated in Table 5, the predicting ability of the enhanced model was highly improved from 86.71% to 95.17%. The improved model for predicting diesel removal efficiency from all significant soil washing terms with removing non-significant interaction terms is presented in Eq. (2).

$$
\begin{aligned}
\text{Diesel removal efficiency (\%)} =\ &71.7 + 3.2 * A + 3.5 * B + 3.0 * C + 4.2 * D \\
&+ 5.1 * E - 0.8 * AE + 0.8 * BD + 0.8 * CE \\
&- 0.8 * DE - 2.2 * A^2 - 1.4 * B^2 - 1.9 * C^2 \\
&- 3.2 * D^2 - 2.5 * E^2
\end{aligned}
$$

## Removal efficiency

Response surface plots or contour plots generated by RSM could be used to explain the interaction between variables and to find the variables optimum values (*Xu et al., 2017*).

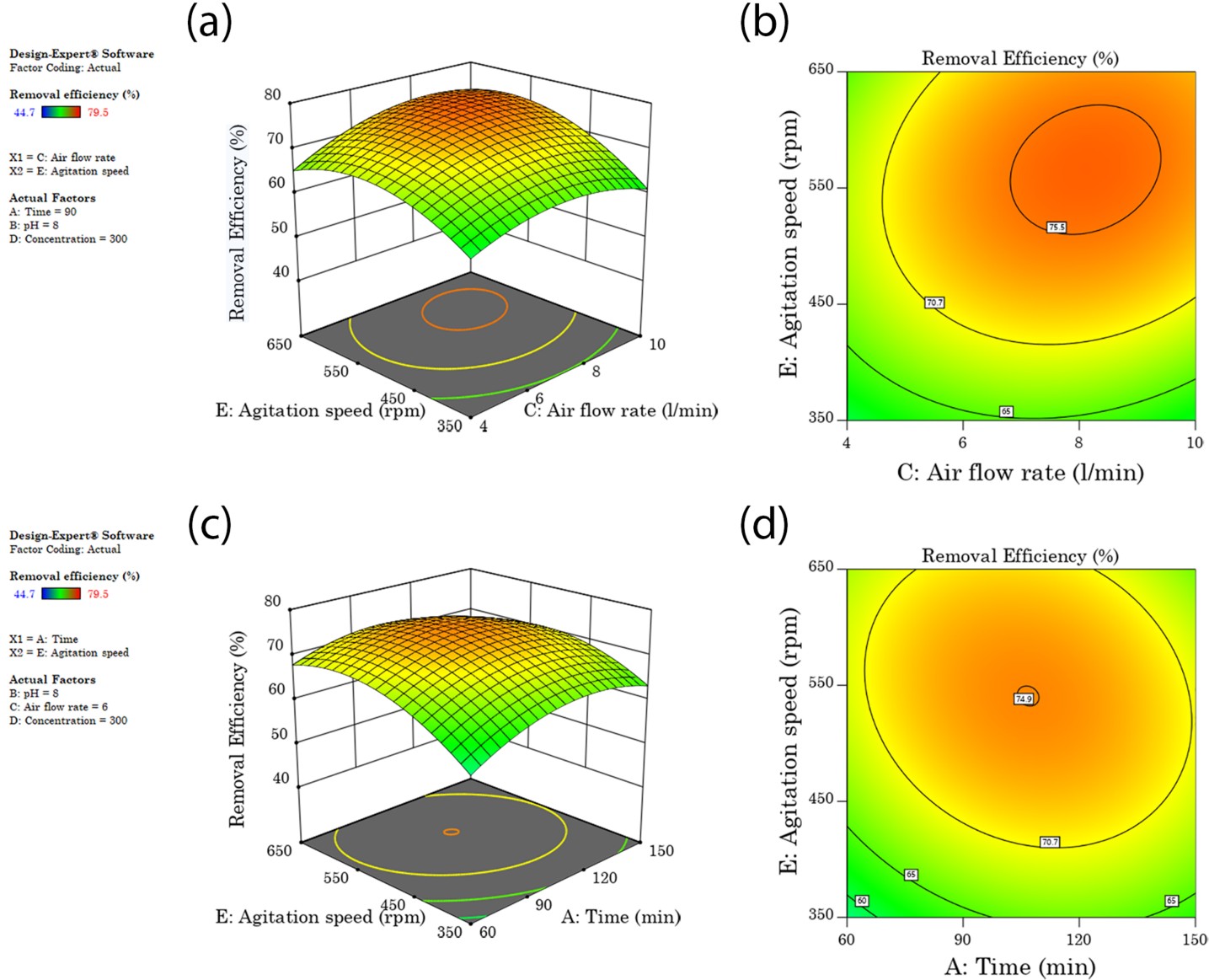

**Figure 3 Surface and contour plots of significant interaction terms (A and B) of air flow rate and agitation speed interaction surface and contour plot of time and agitation speed (C and D).**

The response surface plot for diesel removal efficiency as affected by the five washing parameters and levels is shown in Figs. 3 and 4. All the tested factors agitation speed, concentration, soil solution pH, washing time, and airflow rate bring a significant effect on diesel removal efficiency. When there are significant interaction terms in the model, it is not correct to interpret the individual effect of each factor as their impact is affected by the level of the other factors (*Montgomery, 2013*). From the ten combinations of interaction factors, four combinations were found to have a significant effect on the model, airflow rate and agitation, time and agitation, concentration and agitation, and concentration and pH showed significant interaction as shown in Table 6.

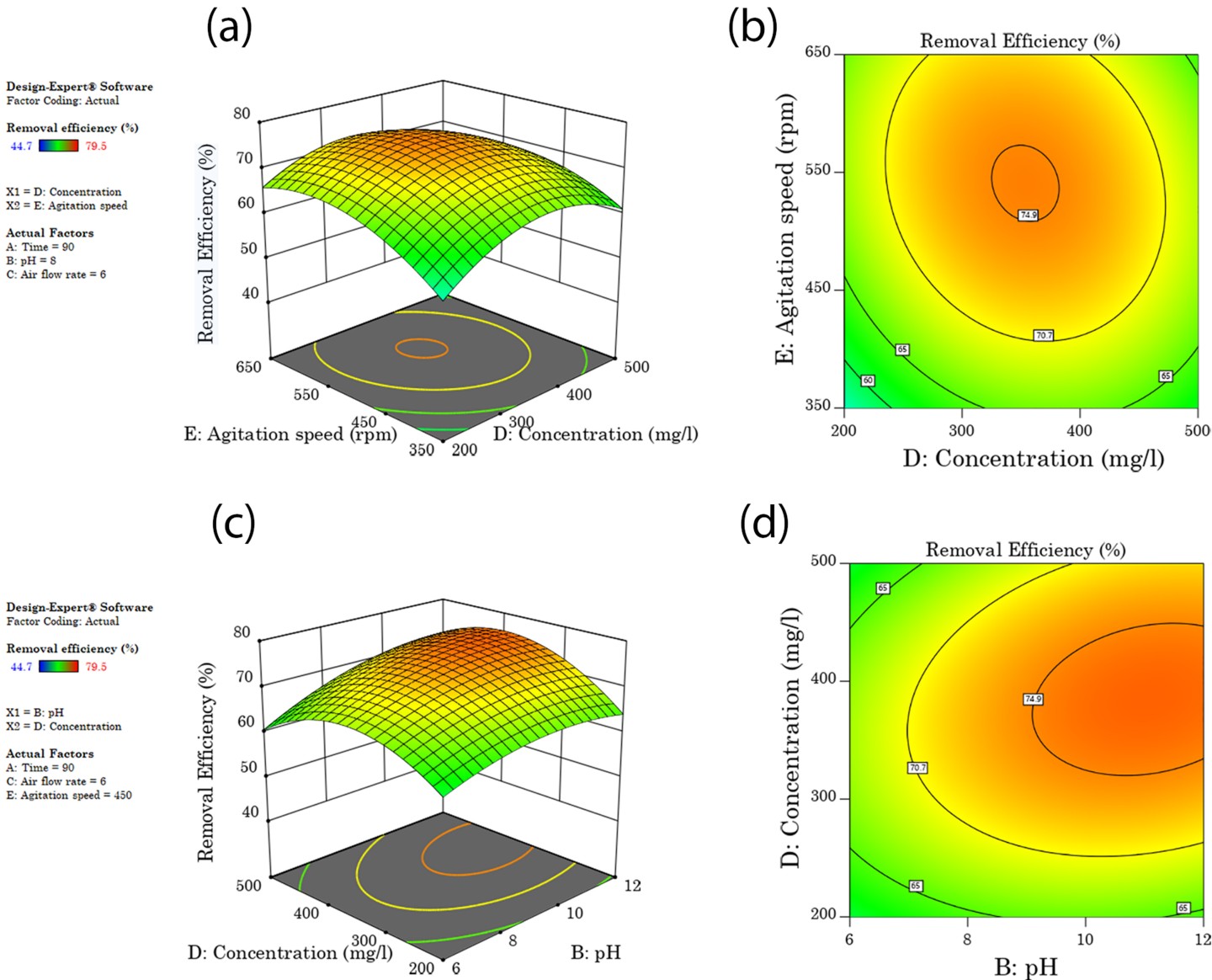

**Figure 4 Interaction surface and contour plot of agitation speed and surfactant concentration (A and B), and (C and D) interaction surface and contour plots of concentration and washing pH.**

### Airflow rate and agitation speed interaction effect

The interactive effects between airflow rate and agitation speed on diesel removal efficiency, keeping the other three factors constant at the central point are shown on the 3D response surface and contour plot in Figs. 3A and 3B. An increase in both terms brought a rise in diesel removal efficiency, and optimum diesel removal efficiency of 75.5% was recorded when both airflow rate and agitation speeds were in a range of 6.8–9.2 l/min and 520–610 rpm respectively. Further increase or decrease in airflow rate and agitation speed from this range brought a reduction in diesel removal efficiency as shown in Figs. 3A and 3B. As airflow rate increases more hydrophobic microbubbles will be created with the large interfacial surface area and high negative zeta potential, which will enhance the

**Table 6 Analysis of variance table.**

| Source | Sum of squares | df | Mean square | F Value | p-Value Prob > F | |
|---|---|---|---|---|---|---|
| Model | 2451.08 | 20 | 122.55 | 65.57 | <0.0001 | Significant |
| A-Time | 240.67 | 1 | 240.67 | 128.76 | <0.0001 | |
| B-pH | 288.43 | 1 | 288.43 | 154.31 | <0.0001 | |
| C-aeration | 220.83 | 1 | 220.83 | 118.15 | <0.0001 | |
| D-concentration | 420.01 | 1 | 420.01 | 224.71 | <0.0001 | |
| E-agitation | 614.08 | 1 | 614.08 | 328.54 | <0.0001 | |
| AB | 3.61 | 1 | 3.61 | 1.93 | 0.1921 | |
| AC | 0.81 | 1 | 0.81 | 0.43 | 0.5239 | |
| AD | 1.21 | 1 | 1.21 | 0.65 | 0.4381 | |
| AE | 10.24 | 1 | 10.24 | 5.48 | 0.0391 | |
| BC | 1.32 | 1 | 1.32 | 0.71 | 0.4182 | |
| BD | 11.22 | 1 | 11.22 | 6.00 | 0.0322 | |
| BE | 0.72 | 1 | 0.72 | 0.39 | 0.5468 | |
| CD | 2.72 | 1 | 2.72 | 1.46 | 0.2528 | |
| CE | 9.92 | 1 | 9.92 | 5.31 | 0.0417 | |
| DE | 10.56 | 1 | 10.56 | 5.65 | 0.0367 | |
| A2 | 140.80 | 1 | 140.80 | 75.33 | <0.0001 | |
| B2 | 53.73 | 1 | 53.73 | 28.75 | 0.0002 | |
| C2 | 103.50 | 1 | 103.50 | 55.37 | <0.0001 | |
| D2 | 305.73 | 1 | 305.73 | 163.57 | <0.0001 | |
| E2 | 189.38 | 1 | 189.38 | 101.32 | <0.0001 | |
| Residual | 20.56 | 11 | 1.87 | | | |
| Lack of fit | 11.79 | 6 | 1.97 | 1.12 | 0.4600 | Not significant |
| Pure error | 8.77 | 5 | 1.75 | | | |
| Cor total | 2,471.64 | 31 | | | | |

Note:
R-Squared 0.9917, Adj R-Squared 0.9766, Pred R-Squared 0.8671 and Adeq Precision 30.27.

contact surface area between the pollutants and the surfactant solution that help them to detach pollutants (*Agarwal & Liu, 2017*). The detachment of contaminants is due to reduced interfacial tension and bubble collapse forces created due to collusion with soil particles and the shear stress attributed to fluid velocity and subsequent lifting of contaminants to the surface of the washing container (*Kim, Kim & Han, 2012*). Incorporation of aeration to the soil washing system will also help in exposing the volatile and semi-volatile organic compounds for volatilization (*Chao, Hsieh & Tran, 2018*; *Chao, Ong & Huang, 2008*).

### Washing time and Agitation speed interaction effect

Washing time and Agitation speed showed a significant interaction effect in the model, as shown in Table 6. Figures 3C and 3D shows the combined effect of washing time with agitation speed, increasing washing time and agitation speed simultaneously brought an increase in diesel removal efficiency and reached an optimum diesel removal efficiency

of 74.9% when washing time and agitation speeds were increased to a range of 93–118 min and 500–580 rpm respectively. A further increase or decrease in washing time and agitation speed from this range brought a reduction in diesel removal efficiency. An increase in washing time will increase the contact time between the surfactant solution and the contaminant which favors micellization and solubilization of petroleum contaminants from the soil colloid surface and joining the washing solution, but after a certain period equilibration will be attained (*Baziar et al., 2013*; *Khalladi et al., 2009*). An increase in washing time is associated with an increase in the volume of treatment container, or a reduction in treatment capacity at a given time (*Zhang & Lo, 2007*).

### Agitation speed interaction effects

Figures 3A–3D, 4A and 4B shows the combined effect of agitation speed with airflow rate, washing time and concentration, respectively. Diesel removal efficiency first increased with an increase in agitation speed and then leveled off. This was because of the higher shear force used under high agitation speed that will help in proper mixing and enhance the contact area between the pollutant and the surfactant solution. When agitation speed increases the collusion between soil particles increases, which helps for striping-off of the contaminants from the soil colloid surfaces or makes the weakly bound contaminants susceptible for easier removal by surfactants. A higher agitation speed has also been reported to favor micelle formation (*Urum, Pekdemir & Copur, 2004*). The optimum diesel removal efficiency of 74.5–76.3% was attained when agitation speed was increased to the range of 500–630 rpm and with a simultaneous increase of airflow rate, washing time and surfactant concentration to 6.5–9.5 l/min, 93–118 min and 320–480 mg/l respectively. After a certain level of higher agitation speed, the soil will start moving as a unit, and a further increase in agitation speed may not bring extra increment in efficiency (*Baziar et al., 2013*).

### Soil pH and surfactant interaction effect

Figures 4C and 4D shows the combined effect of soil pH and surfactant concentration; diesel removal efficiency increased when both pH and concentration were increased to 9 and 320–440 mg/l respectively. An optimum diesel removal efficiency of 75% was reached at the mentioned pH and concentration ranges, and further increase or decrease in concentration brought a decrease in diesel removal efficiency. An increase in pH of the soil solution increases the negative charges of both organic pollutants and the inorganic soil colloid surfaces. This increase in negative charge will result in dispersion of soil particles and desorption of organic pollutants from the colloid surface to the washing solution, due to repulsion between negatively charged heads of organic matter and soil colloids (*Youa, Yinb & Allen, 1999*; *Zhang & Lo, 2007*). As soil solution pH increases the solubility of organic matter also increases as the essential components of organic matter (humic acid and fulvic acid) are highly soluble in alkaline solutions (*Curtin, Peterson & Anderson, 2016*). When the organic matter solubilizes into the soil solution, the highly adsorbed petroleum hydrocarbon molecules are also released or exposed for solubilization by surfactants through reduction of surface tension. On a column study conducted on diesel

**Table 7 Optimization table.**

| Factors and levels | | | | | | Predicted removal efficiency (%) | Actual removal efficiency (%) | Error | Desirability |
|---|---|---|---|---|---|---|---|---|---|
| Exp No. | Time (Min) | pH | Airflow rate (l/min) | Concentration (mg/l) | Agitation speed (rpm) | | | | |
| 1 | 90 | 10 | 7.2 | 362 | 537 | 79.5 | 78.3 | 1.2 | 1 |
| 2 | 71 | 10 | 8 | 362 | 550 | 77.8 | 78.3 | 0.5 | 0.82 |
| 3 | 70.5 | 10 | 8 | 367 | 550 | 77.7 | 76.8 | 0.9 | 0.82 |

contaminated soils, *Salehian, Khodadadi & Hosseini (2012)* reported in an increase in pH or alkaline phase more diesel was extracted than an acidic phase and claimed that oils are more soluble under alkaline condition than acidic.

### Concentration interaction effect

Figures 4A–4D show the combined effect of concentration with agitation speed and pH respectively. Diesel removal efficiency increased when surfactant concentration increased to a range of 320–440 mg/l and then leveled off. An optimum diesel removal efficiency of 75% was attained at the forgoing concentration ranges. At lower concentrations usually at a concentration less than their CMC, surfactant molecules tend to sorb on soil colloids when their concentration increases the adsorption sites will be full and surfactant molecules will start to form micelles and then solubilize contaminants (*Zhang & Lo, 2007*). The increase in surfactant concentration above CMC provides excess surfactant molecules for micellization, which ultimately increase the mobilization and solubilization of pollutants from the soil colloid to the washing solution by reduction of interfacial tension (*Villa, Trovó & Nogueira, 2010*).

### Optimization and validation using desirability approach in RSM

The optimum operating conditions for the five soil washing parameters were assessed with desirability approach using Design Expert 10 software, with a goal of finding higher diesel removal efficiency. Based on desirability assessment in the experimental range (−1 to +1 in coded values) three optimization conditions with desirability value of 0.82–1.00 were selected as shown in Table 7, with a predicted diesel removal efficiency of 77.7% to 79.5%. Validation tests with three replications were conducted for the three types of optimized values, and the variation in predicted and experimental results for all the tests was in the range of +1.2%. This result confirms that the model developed has a good predicting ability and could be used for future soil washing studies conducted on the current soil studies. To improve the predicting ability of this model under different soils further calibration studies should be conducted and with more data's this predictor could be developed into a full model. The results of the predicted and the experimented results are presented in Table 4. The final diesel concentration in the soil after the optimized washing condition was 11.275 gm/kg with a diesel removal of 43.725 gm/kg from the initial diesel concentration of 55.5 gm/kg of diesel. Because the initial diesel concentration was very high and the soil washing was conducted in the finer fraction of the soil (<0.841 mm) even after around 80% removal it did not reach below the regulatory standard. According

to the current regulatory standard of China, a total petroleum hydrocarbon of <3 gm/kg is allowed for agricultural use (*Luo et al., 2019b*) to meet the regulatory standard a secondary sequential washing was undertaken which gave a final diesel concentration of <2.367 g/kg of soil. Similar results of lower diesel removal rate due to high initial diesel concentration in the soil are reported by *Streche et al. (2018)*.

## Effect of soil aeration on different organic matter soils

Even though the soils were contaminated with the same 10% diesel oil, an increase in organic matter levels was associated with minimal diesel loss due to weathering. The soil with low organic matter addition relatively loses more diesel due to evaporation than the one with higher. The main reason for this case is, as organic matter level increases in the soil the hydrophobic interaction between the pollutant and the organic matter will be stronger, and it will be less susceptible to weathering (*Lee, 2010*; *Sui et al., 2014*). As the amount of added organic matter increases from 0% to 7% the diesel removal efficiency of aerated assisted soil washing and normal soil washing shows a slow decrease in diesel removal efficiency in all four surfactants tested as shown in Figs. 5A and 5B. Higher organic matter levels will give high surface area, will hold the contaminants strongly due to hydrophobic attraction and give strong hydrophobicity to the soil surface which will make it difficult for surfactant solutions to remove the pollutants (*Sui et al., 2014*; *Zhou & Zhu, 2007*). As organic matter content increases the difference in diesel removal efficiency gets wider; without organic matter addition the difference in removal was 12.2–14.3% but when organic matter increases to 7%, the difference grows to 16–18.1%. This implies aerated assisted soil washing performs better under both high and low organic matter levels. The order of diesel removal efficiency under different surfactant conditions was in the order of Brij 35 > TX100 > Tween 80 > Saponin as shown in Figs. 5A and 5B.

## Effect of aeration on soil textural classes

The performance of aerated assisted and regular soil washing was assessed under four particle size classes as shown in Figs. 5C and 5D. The diesel removal trend from the four classes of soil texture reveals; more diesel was removed from coarse-textured soils as compared to finely textured classes in both washing methods. As the micro and mesopores of fine soil particles are smaller, they will have closer contact with the contaminant and will be held firmly by hydrophobic attraction. Whereas, coarser particles have smaller surface area,lower organic matter content and cation exchange capacity than fine particles which will lightly held the contaminants than finer soil particles (*Amirianshoja et al., 2013*; *Chao, Hsieh & Tran, 2018*). The higher surface area of finer particles will also affect pollutant removal as the surfactant absorption capacity of the soils will increase which ultimately affecting the effectiveness of soil washing.

When we compare the removal efficiency of aeration assisted soil washing, and regular surfactant-assisted soil washing in the same textural classes, as particle size gets smaller the difference in efficiency gets wider. When the soil particles are larger, the difference in removal efficiency between the two methods is around 10.2–13%, but when particle size gets smaller, the difference grows to 20.2–26.5%. These findings indicate that for

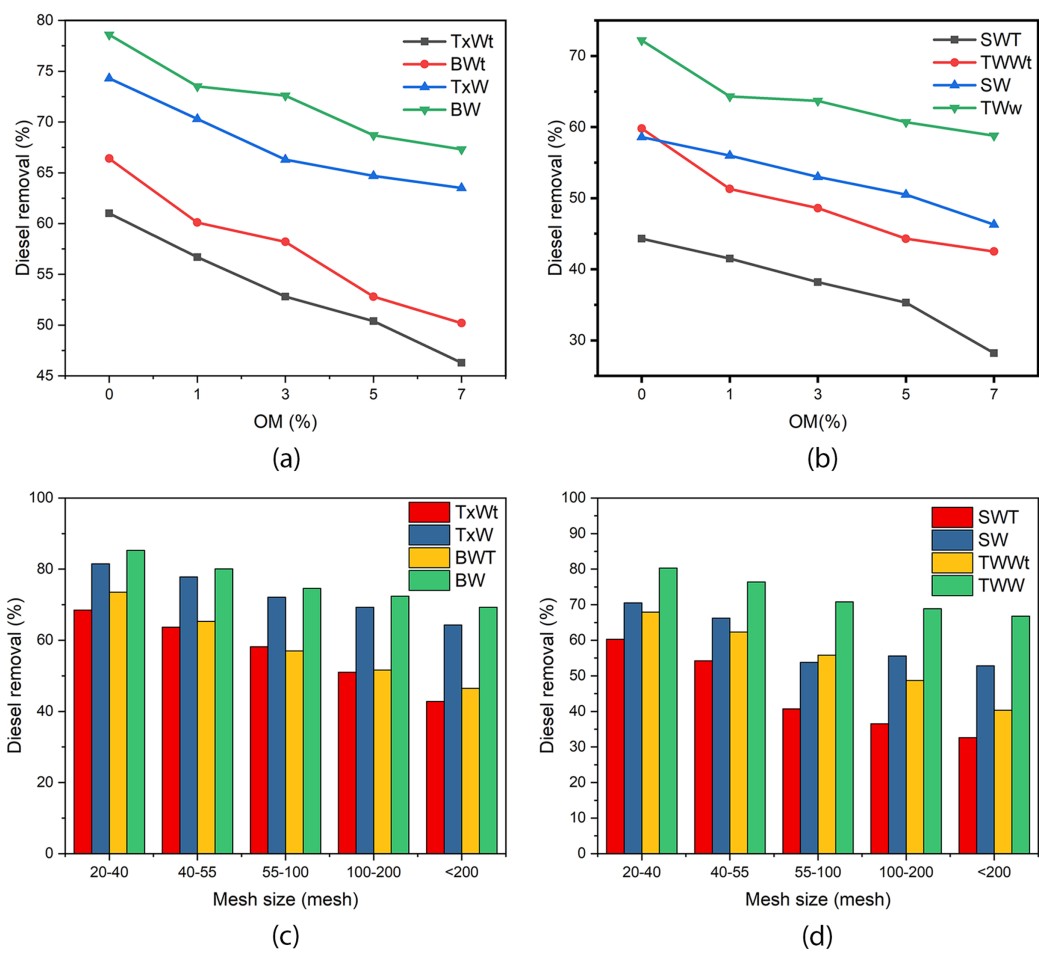

**Figure 5 (A) and (B) are soil washing efficiency under different organic matter, (C) and (D) are soil washing efficiency under different particle sizes.** BW and BWt represent Brij 35 with and without aeration respectively, TxW and TxWt are TX100 with and without aeration, TWW and TWWt are Tween 80 with and without aeration, SW and SWt are saponins with and without aeration respectively.

larger particle sizes or sandy soils regular soil washing is efficient but for finer or clay soils aerated assisted soil washing should be a preferred method. This finding shows if we want to use aeration assisted soil washing we have to make size separation at the beginning after that wash the large size particles in regular surfactant-assisted soil washing and the finer particles should be addressed independently with aeration and surfactant-enhanced soil washing. This finding indicates that it is wise to use aerated assisted surfactant enhanced soil washing for smaller soil particle sizes to maximize its better efficiency. The diesel removal trend in the four surfactant classes and particle size classes follows similar trend Brij 35 > TX100 > Tween 80 > Saponin as shown in Figs. 5C and 5D.

## Effect of aeration on the aging of contaminants

The concentration of diesel in artificially contaminated soil decreased from initial 100 mg/gm of soil diesel concentration to 75, 68, 61.3, 49.7 and 46.5 mg/g during 2, 5, 9

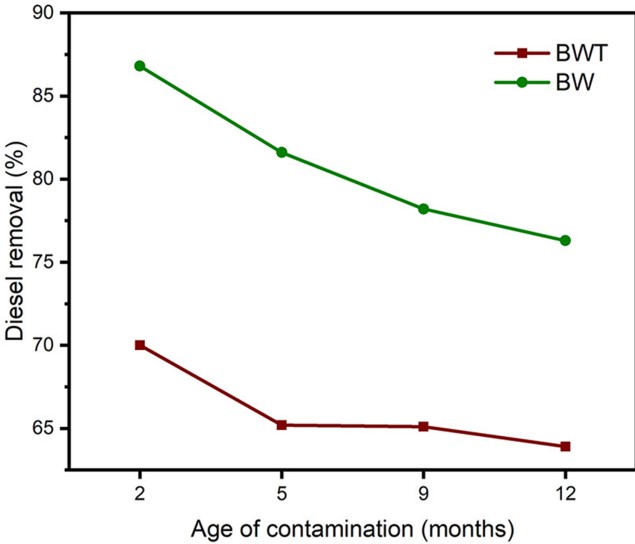

**Figure 6** The effect of age of contamination on diesel removal efficiency; BWT and BW are Brij 35 with and without aeration respectively.

and 12 months of contamination respectively, implying with aging light volatile diesel compounds were lost due to evaporation. Aging of contaminated soils affects the removal efficiency of treatment methods, as the contaminant gets aged it is subjected to different environmental conditions, and the light volatile compounds will be removed by evaporation, and the readily degradable components will degrade and contaminant components with higher density and hard to remove stays in the soil (*Uhmann & Aspray, 2012*; *Zhang & Lo, 2007*). Both soil-washing methods show a decrease in removal efficiency as the age of contamination increases. Different researchers report that the more contaminants are available for surfactant solubilization, the higher the removal efficiency and vise versa due to the removal of light volatile fractions by evaporation and degradation (*Chen, Xing & Xie, 2007*). The difference in removal efficiency between the two methods was relatively constant until five months with a relative difference of 16.6% but when aging of contaminants increase the difference closes to 12.4% at 12 months of contamination as shown in Fig. 6. These results bring us to concluding aeration assisted soil washing is effective for soils contaminated for both short and extended periods, and it is highly effective for remediation of newly contaminated soils.

## CONCLUSIONS

The effect of incorporation of microbubbles in the ex-situ soil washing process was assessed, and it was found that aeration increased diesel removal efficiency from 12% to 25% depending on particle size, organic matter levels and age of contamination. RSM with the CCD was effective in identifying interacting factors and optimizing the washing parameters. The predictor model developed was able to predict the washing efficiency with a +2.2% efficiency. This aeration assisted soil washing was found to be effective for fine particle size and high organic matter containing soils as compared to non-aerated soil washing, this is a promising result as soil washing is reported to be less effective under both

conditions. The findings of this article depicted RSM was effectiveness in modeling, optimizing and predicting soil washing removal and incorporation of aeration reduces the concentration of surfactant required to remove the pollutants found in the soil.

## ACKNOWLEDGEMENTS

The authors greatly acknowledge the support of the college of environmental science and engineering department of Donghua University for supplying lab facilities.

### Funding
This work was financially supported by the National Key Research and Development Program of China (Contract Numbers 2016YFC0400501 and 2016YFC0400502) and the National Natural Science Foundation of China (Contract Number 21277023). The funders had no role in study design, data collection and analysis, decision to publish, or preparation of the manuscript.

### Grant Disclosures
The following grant information was disclosed by the authors:
National Key Research and Development Program of China: 2016YFC0400501 and 2016YFC0400502.
National Natural Science Foundation of China: 21277023.

### Competing Interests
The authors declare that they have no competing interests.

### Author Contributions
- Befkadu Abayneh Ayele conceived and designed the experiments, performed the experiments, analyzed the data, prepared figures and/or tables, authored or reviewed drafts of the paper, and approved the final draft.
- Jun Lu performed the experiments, analyzed the data, prepared figures and/or tables, authored or reviewed drafts of the paper, and approved the final draft.
- Quanyuan Chen conceived and designed the experiments, authored or reviewed drafts of the paper, and approved the final draft.

### Field Study Permissions
The following information was supplied relating to field study approvals (i.e., approving body and any reference numbers):

Mr. Zheng Sun the owner of the farm in Songjiang district, Shanghai and approved soil sample collection at the study site.

### Data Availability
The raw and supplementary data is available in a Supplemental File.

## Supplemental Information

Supplemental information for this article can be found online at http://dx.doi.org/10.7717/peerj.8578#supplemental-information.

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
