# Peer review of "Optimization of aeration enhanced surfactant soil washing for remediation of diesel-contaminated soils using response surface methodology"

_PeerJ, doi:10.7717/peerj.8578_

## Round 0.1 · original submission · Minor Revisions

Please pay careful attention to the reviewer comments and revise your manuscript accordingly.

·

Basic reporting

The author made nice effort to improved the soil soil washing performance by 12-25% through incorporation of air bubbles into the low concentration surfactant soil washing system.

Experimental design

Although the experimental process was completed in about one year starting from soil sampling to their drying and grinding and then contamination with diesel, but written well and can be understand easily.

Validity of the findings

The results shows interesting improvement in the remediation of diesel contaminants from soil. These results are interesting, but i suggest to improve the validity of this technique, to use this technique on large scale as the soil of the near by areas to oil refineries, processing plants and oil field are highly contaminated with hydrocarbons.

Additional comments

I read with much interest this manuscript reporting on " Optimization of aeration enhanced surfactant soil washing for remediation of diesel-contaminated soils using Response Surface Methodology". Looking to the environmental issues related to the soil contamination. The author made nice effort to improved the soil soil washing performance by 12-25% through incorporation of air bubbles into the low concentration surfactant soil washing system. Overall, I found the manuscript interesting, with certain novelty over the existing state of the art. Based on the details given in the manuscript, I believe it can be improved still and can be brought up to meet the expectations from a paper in Peer J. Besides, there are some issues which need to be answered by the authors in this version of the manuscript. Also, state of art needs to be defined better and some key references are missing. The language also needs careful revisions. Thus, the manuscript as it is written now, the work needs minor revisions. Any other comments for improving the manuscript are-
1. Although Figure 1 explains the soil washing scheme, but that is mentioned as Fig 1. He manuscript quality can be improved by adding suitable graphical abstract.
2. Introduction: The introduction of the manuscript looking long. Concise the text to bring the impact of the need.
3. In the introduction part on page 2 the wrote that “Different physical, chemical, and biological remediation techniques are commonly used to treat the petroleum polluted soils; these methods aim at destroying, modify them into harmless byproducts and reducing contaminant concentration”. Authors may check this recent paper to strengthen this aspect- 2019, Environmental perspectives of interfacially active and magnetically recoverable composite materials–A review Science of the Total Environment 670 (2019) 523–538.
4. It is suggested to insert more relevant recent reference citations at appropriate instances for better readership.
5. The grammar of the article should be re-checked. There are many mistakes. English grammar must be improved, especially concordance between subjects and verbs.

·

Basic reporting

This manuscript integrated the additional physical mixer (i.e., air bubble) with the conventional contaminated soil washing using surfactant solution. The quality of manuscript is within standard both English and design of experiment. The outcome also presented the possibility of this technology in the real application.

Experimental design

This manuscript could be divided into 3 sections which were surfactant selection, process optimization by CCD and technology comparison (with and without aeration). In my opinion, the author should have additional figure which showed the overall of this work. This should make the reader easily understand the flow of this manuscript.
CCD is the useful RSM since it could reduce the number of experiment. However, this manuscript used a large number of factors (time, pH, air flow rate, surfactant concentration and agitation) which might not be suitable for CCD and very hard to explain the results. It should have a screen experiment for important factor and its boundary prior to CCD such as sequential experiment or Taguchi method. Did the author conduct the preliminary study?
In this research field, liquid to solid ratio is very important factor. Please explain why the author did not consider this factor. The author used 20 mL/g as the fixed liquid to solid ratio. Several works attempt to reduce liquid to solid ratio due to the cost of operation (i.e., surfactant and infrastructure). Generally, they operated between 3-10 mL/g.

Validity of the findings

Did the author conduct the replication? If so, please add standard deviation in the revised manuscript.

Additional comments

- There were several mistake according to space between number and unit (Line 173, Line 175). Please check the whole document the correct this issue.
- In table 3 please check the unit: Min vs min, Mg, mg
- Applying aeration improved the diesel removal and surfactant usage. Did it economic feasible? The author mention about applying temperature in soil washing was uneconomical (Line 228).
- I would like to encourage the author for soil characterization, especially soil texture (composition of sand, silt and clay). This soil texture significantly affected to the diesel removal performance.
- The author used UV spectrophotometer at 225 nm to evaluate the diesel concentration (Line 175) Please add the reference. Diesel is the group of long chain hydrocarbon compounds not a single chemical. Did the simple UV spectrophotometer provide the good results? Is there any interfere from dissolved organic matter in soil?
- The author should report the final diesel concentration in washed soil and compared it to the standard.

---

## Round 0.2 · accepted · Accept

Thank you for your efforts to revise your manuscript based on the reviewer comments.

·

Basic reporting

no comment

Experimental design

no comment

Validity of the findings

no comment

Additional comments

I am appreciated to your effort in revising this manuscript.